# VISUAL AUTOREGRESSIVE TRANSFORMERS MUST USE $\Omega(n^2d)$ MEMORY

## ABSTRACT

A fundamental challenge in Visual Autoregressive models is the substantial memory overhead required during inference to store previously generated representations. Despite various attempts to mitigate this issue through compression techniques, prior works have not explicitly formalized the problem of KV-cache compression in this context. In this work, we take the first step in formally defining the KV-cache compression problem for Visual Autoregressive transformers. We then establish a fundamental negative result, proving that any mechanism for sequential visual token generation under attention-based architectures must use at least $\Omega(n^2d)$ memory, when $d = \Omega(\log n)$, where $n$ is the number of tokens generated and $d$ is the embedding dimensionality. This result demonstrates that achieving truly sub-quadratic memory usage is impossible without additional structural constraints. Our proof is constructed via a reduction from a computational lower bound problem, leveraging randomized embedding techniques inspired by dimensionality reduction principles. Finally, we discuss how sparsity priors on visual representations can influence memory efficiency, presenting both impossibility results and potential directions for mitigating memory overhead.

## 1 INTRODUCTION

Visual generation technologies have become integral to a wide range of applications, spanning image enhancement (Lin et al., 2025; Guo et al., 2025), augmented reality (Azad et al., 2024b), medical diagnostics (Azad et al., 2024a; Ma et al., 2024a; Li et al., 2024), and creative fields like game development (Rafner et al., 2020; Chen et al., 2025). By converting text descriptions and other inputs into rich, diverse visuals, these models are transforming both image interpretation and content creation. Key approaches in this domain include Variational AutoEncoders (VAE) (Doersch, 2016), Generative Adversarial Networks (GAN) (Goodfellow et al., 2020), and Diffusion models (Ho et al., 2020). Their ability to generate high-resolution, realistic, and varied imagery has expanded the possibilities of visual synthesis, enhancing fidelity, diversity, and overall quality.

Among these, Visual Autoregressive models have demonstrated remarkable generative capabilities, particularly in sequential token-based image synthesis. However, a fundamental challenge remains: the substantial memory overhead required to store previously generated representations during inference. Existing efforts to alleviate this issue largely rely on compression techniques, yet none have explicitly formalized the problem of KV-cache compression in the context of Visual Autoregressive transformers. One particularly compelling variant is the Visual AutoRegressive (VAR) Transformer (Tian et al., 2025) which adapts the transformer paradigm to structured image synthesis. By employing a coarse-to-fine "next-scale prediction" approach, VAR Transformers produce high-quality images more efficiently than many standard diffusion-based methods (Song et al., 2020).

In this work, we aim to take the first step in rigorously defining KV-cache compression for such models and establish a fundamental lower bound on memory consumption.

**Problem 1.1.** *Is it possible to improve (Tian et al., 2025) to enhance the compression cost of KV-Cache on Visual Autoregressive Transformers?*

To Answer Problem 1.1, we first review some previous results of Visual Autoregressive Transformers in (Ma et al., 2024b; Sun et al., 2024; Tian et al., 2025; Ke et al., 2025) and then formalize the KV-Cache compression problem in Visual Autoregressive Transformers.

**Definition 1.2** (KV Cache Compress Problem in VAR Transformer, informal version of Definition 3.7). *At the $i$-th iteration of* VAR *transformer, the model will output the $i$-th scale of feature map $Z_i$ with size $h_i \times w_i$ (See Definition 3.6). In this process, the newly input is a $h_i w_i \times d$-dimension matrix triple $(Q_i, K_i, V_i) \in (\mathbb{R}^{d \times (h_i w_i)})^3$. After the triple entry, the Attention function computes:*

$$\mathsf{Attn}(Q_i, \mathsf{K}_i, \mathsf{V}_i) := \sigma_i (\mathsf{K}_i \cdot Q_i)^\top \cdot \mathsf{V}_i \in \mathbb{R}^{(h_i w_i) \times d}$$

*where:*

$$\mathsf{K}_i = \begin{bmatrix} K_1^\top \\ K_2^\top \\ \vdots \\ K_i^\top \end{bmatrix}, \quad \mathsf{V}_i = \begin{bmatrix} V_1^T \\ V_2^T \\ \vdots \\ V_i^T \end{bmatrix}$$

*are $\sum_{j=1}^{i} (h_j w_j) \times d$ matrices, and $\sigma_i : \mathbb{R}^{\sum_{j=1}^{i} (h_j w_j) \times h_i w_i} \to \mathbb{R}^{\sum_{j=1}^{i} (h_j w_j) \times h_i w_i}$ is the softmax function for each column. The $\mathsf{K}_i$ and $\mathsf{V}_i$ matrices are called the key-value (KV) cache.*

To the best of our knowledge, there are no formal results to support and describe such approaches in a comprehensive fashion. To bridge this gap, we propose the following question and develop a foundational theory to characterize the complexity of KV-cache compression problem for Visual Autoregressive Transformers.

**Problem 1.3.** *What is the space complexity lower bound for KV-Cache in Visual Autoregressive transformer?*

In this work, we answer both Problem 1.1 and Problem 1.3, we prove that any mechanism for sequential visual token generation under attention-based architectures must use at least $\Omega(n^2 d)$ memory, when $d = \Omega(\log n)$, where $n$ represents the number of generated tokens and $d$ denotes the embedding dimensionality.

**Theorem 1.4** (Space Complexity Lower Bounds for Key-Value Cache in Precise Attention Computation, informal version of Theorem 4.5). *Let $N$ denote the total number of feature map scales generated by* VAR *transformer, where the height and width at the $N$-th level satisfy $h_N = w_N = n$. There exists some universal constant $C_u > 1$ and for $d \geq C_u \log n$, any algorithm that can, with probability at least $\frac{9}{10}$ produce an output*

$$o_N := \mathsf{Attn}(Q_N, \mathsf{K}_N, \mathsf{V}_N) \in \mathbb{R}^{n^2 \times d}$$

*must use at least $\Omega(n^2 d)$ bits of memory.*

This result indicates that achieving truly sub-quadratic memory usage is infeasible without imposing additional structural constraints. To support our findings, we construct a proof via reduction from a computational lower-bound problem, leveraging randomized embedding techniques inspired by dimensionality reduction principles. Furthermore, we explore the role of sparsity priors in visual representations, discussing both theoretical impossibility results and potential avenues for mitigating memory overhead. By formally characterizing these constraints, our work provides a foundational step toward designing more memory-efficient Visual Autoregressive models.

## 1.1 OUR CONTRIBUTIONS

We summarize our contributions as follows:

- We formally define the KV-cache compression problem in Visual Autoregressive Transformers and highlight the lack of prior theoretical results in this domain.

- We establish a fundamental space complexity lower bound of $\Omega(n^2 d)$ for KV-cache storage in Visual Autoregressive Transformers, proving that subquadratic memory usage is infeasible without structural constraints.

- We develop a rigorous proof technique leveraging reductions from known computational lower-bound problems, providing a new perspective on memory efficiency in Visual Autoregressive transformers.

## 1.2 ROADMAP

This paper is structured as follows: we begin with a review of related work in Section 2 followed by essential definitions and foundational concepts in Section 3. In Section 4 we present our lower bound result. Finally, we conclude in Section 5.

## 2 RELATED WORK

This section briefly reviews the related research work on VAR, KV-Cache compression and theoretical limitations of transformer. These topics have a close connection to our work.

### 2.1 VISUAL AUTOREGRESSIVE MODELS

Autoregressive modeling (Touvron et al., 2023; Bai et al., 2023; Achiam et al., 2023; Grattafiori et al., 2024; Liu et al., 2024; Ma et al., 2024c; Lu et al., 2024; Kakogeorgiou et al., 2024; Piergiovanni et al., 2024; Bai et al., 2024), rooted in NLP, employs sequential prediction where Transformers are pretrained to generate the next token. This paradigm was extended to vision via PixelCNN (Van den Oord et al., 2016), which modeled pixel-level likelihoods using CNNs, and later VQGAN (Esser et al., 2021; Weber et al., 2024), which integrated autoregression into VQ-VAE's compressed space for tractable probabilistic synthesis. LlamaGen (Sun et al., 2024) further adapted next-token prediction from LLMs to image generation, showing that scaled autoregressive models like Llama (Grattafiori et al., 2024) can reach state-of-the-art performance without explicit visual priors. Recent work (Tian et al., 2025) advanced from token-wise to scale-wise autoregression, introducing coarse-to-fine next-scale prediction that surpasses diffusion models in scalability, speed, and quality. Our analysis of KV-cache compression in Visual Autoregressive models (Tian et al., 2025) provides new heuristic insights for this line of research.

### 2.2 KV CACHE COMPRESSION

Recent work has focused on compressing the KV cache in Transformer LLMs by exploiting structural or learned sparsity. (Zhang et al., 2023) proposed an eviction method via dynamic submodular maximization, while (Xiao et al., 2023) identified "attention sinks"—early tokens with persistently high weights—suggesting a strategy of retaining these sinks with a sliding window of recent tokens. (Liu et al., 2023a) formalized the "persistence of importance" in dominant tokens, and (Liu et al., 2023b) highlighted the role of contextual sparsity. Beyond sparsity, (Zandieh et al., 2024) leveraged clustering patterns in key representations for memory-efficient algorithms with theoretical space advantages. More recently, (Haris & Onak, 2025) studied the lower bounds of KV-cache compression in autoregressive Transformers. Building on this, we extend the theoretical framework to Visual Autoregressive Transformers, offering the first systematic analysis of their compression challenges and novel theoretical insights.

## 3 PRELIMINARY

In this section, we present some preliminary concepts and definitions of our paper. We first introduce the traditional KV-cache compression problem in Section 3.1, then we provide the formalized preliminaries about Visual Autoregressive transformer in Section 3.2. We formalized the KV-cache compression problem in Visual Autoregressive transformers in Section 3.3.

### 3.1 KV-CACHE COMPRESSION PROBLEM

Here we introduce the traditional KV-Cache compression problem.

**Definition 3.1** (KV Cache Compression Problem). *The input is a stream of triples* $(q_i, k_i, v_i) \in (\mathbb{R}^d)^3$, *where* $d$ *is the embedding dimension. After each stream entry, the Attention function is defined as:*

$$\text{Attn}(q_i, K_i, V_i) := \sigma_i(K_i \cdot q_i)^T \cdot V_i \in \mathbb{R}^d,$$

where $K_i = \begin{bmatrix} k_1^\top & k_2^\top & \cdots & k_i^\top \end{bmatrix}, V_i = \begin{bmatrix} v_1^\top & v_2^\top & \cdots & v_i^\top \end{bmatrix}^\top$ are $i \times d$ matrices, and $\sigma_i : \mathbb{R}^i \to \mathbb{R}^i$ is the softmax function with support $[i]$. The $K_i$ and $V_i$ matrices are called the key-value (KV) cache.

**Remark 3.2.** *The attention function can also be viewed as a collection of expected values under suitable softmax distributions. Let $D_i$ be the softmax distribution over $[n]$ corresponding to the values $q_i^T k_1, \ldots, q_i^T k_n$. Then we have*

$$\text{Attn}(q_i, K_i, V_i) = \mathbb{E}_{\ell \sim D_i}[V_\ell].$$

### 3.2 VISUAL AUTOREGRESSIVE TRANSFORMER

In this section, we present the definition of the Visual Autoregressive transformer, which differs in its generation approach from conventional autoregressive transformers traditionally used for text generation. Specifically, the VAR model (Tian et al., 2025) uses the VAR Transformer to convert the initialized token map $X_{\text{init}}$ into a series of pyramid-shaped feature maps, where each scale of the "pyramid" represents a feature map at a different resolution. The VAR Transformer alternates between interpolation layers and attention layers to get the output.

First, we present the up-interpolation operation on a given scale of the feature map.

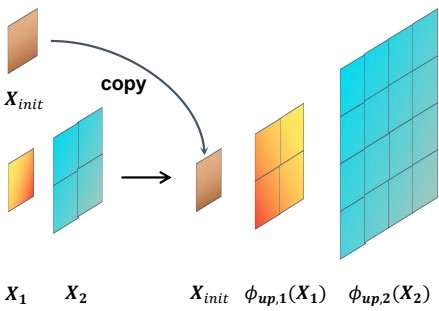

Figure 1: Example of the Pyramid Up-Interpolation Layer $\Phi_{\text{up},2}$ used in the model.

**Definition 3.3** (Up-interpolation Layer). *The layer $\phi_{\text{up},r}$ takes the input feature map $X_r \in \mathbb{R}^{h_r \times w_r \times d}$ and computes the output feature map $Y_{r+1} \in \mathbb{R}^{h_{r+1} \times w_{r+1} \times d}$, where $h_r < h_{r+1}$ are the heights, $w_r < w_{r+1}$ are the widths, and $d \in \mathbb{N}$ is the hidden dimension. Let $U_r \in \mathbb{R}^{h_{r+1}w_{r+1} \times h_r w_r}$ denote a linear transformation matrix. Reshape $X$ into size $h_r w_r \times d$ by flattening its spatial dimensions. The output is defined via:*

$$Y_r = U_r X_r \in \mathbb{R}^{(h_{r+1} w_{r+1}) \times d}$$

*Then reshape back to $Y_r \in \mathbb{R}^{h_{r+1} \times w_{r+1} \times d}$.*

Building on this, we define the pyramid upsampling layer, whose input consists of multi-scale pyramid-shaped token maps.

**Definition 3.4** (Pyramid Up-Interpolation Layer $\Phi$). *The layer $\Phi_{\text{up},k}$ takes the initial token map $X_{\text{init}}$ and the token maps $X_r \in \mathbb{R}^{h_r \times w_r \times c}(r \in [k])$ and computes new token maps $Y_r \in \mathbb{R}^{h_r \times w_r \times c}$. It sets $Y_1 = X_{\text{init}}$ and computes $Y_{r+1} = \phi_{\text{up},r}(X_r)$ as in Definition 3.3. The output is the set consisting of $Y_i(i \in [k+1])$. For clarity, we illustrate the schematic diagram of the $\Phi_{\text{up},2}$ in Figure 1.*

After an up-interpolation layer, the token maps (after being flattened into a proper shape) will be input into an attention layer.

**Definition 3.5** (Single Attention Layer). *Let $X \in \mathbb{R}^{n \times d}$ denote the input matrix. Let $W_Q, W_K, W_V \in \mathbb{R}^{d \times d}$ denote the weight matrix for query, key, and value, respectively. First, compute the attention matrix $A \in \mathbb{R}^{n \times n}$:*

$$A_{i,j} := \exp(X_{i,*} W_Q W_K^\top X_{j,*}^\top), \ \text{for } i, j \in [n].$$

*Then, compute the output:*

$$\mathsf{Attn}(X) := D^{-1}AXW_V,$$

*where* $D := \mathrm{diag}(A\mathbf{1}_n) \in \mathbb{R}^{n \times n}$

Then we are able to define the VAR transformer. A VAR Transformer with $N$ layers alternates between the attention layer and up sample blocks (where the output of each layer is reshaped to a proper shape as the input for the next layer):

**Definition 3.6** (VAR Transformer). *Let $N$ denote the total number of token map scales generated by* VAR *model. The transformer* TF *takes an initial token map $X_{\mathrm{init}} \in \mathbb{R}^{1 \times d}$, computes Part 1. $Z_1 = \mathsf{Attn}_1(X_{\mathrm{init}})$, Part 2. For $k \in \{2, 3, \ldots, K\}$, $Z_k = \mathsf{Last}_{h_k \times w_k}(\mathsf{Attn}_k(\Phi_{\mathrm{up},k-1}(X_{\mathrm{init}}, Z_1, \ldots, Z_{k-2}, Z_{k-1})))$. and finally outputs $\{Z_1, \ldots, Z_N\}$. Here, $\Phi_{\mathrm{up},k}$ is defined in Definition 3.4 and $\mathsf{Attn}_k$ is defined in Definition 3.5. $\Phi_{\mathrm{up},k-1}(\cdot)$ is flatten into shape $\sum_{r=1}^{k}(h_r \times w_r) \times d$ as input for $\mathsf{Attn}_k$. We apply $\mathsf{Last}_{h_k \times w_k}$ to retain the last $h_k w_k \times d$ dimensions of the $\mathsf{Attn}_k$ output and reshape it into $h_k \times w_k \times d$.*

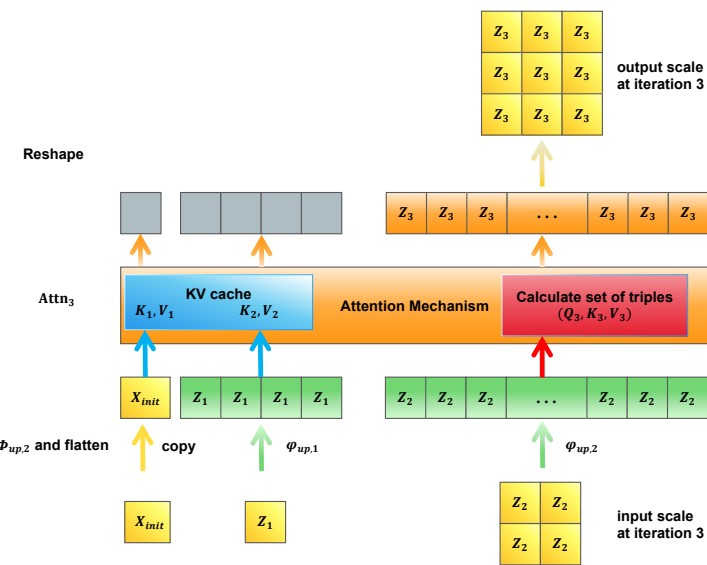

Figure 2: **KV Cache Schematic of the** VAR **Transformer.** We present the example of the 3rd iteration of the VAR Transformer: Before the iteration, the KV cache contains $K_1, V_1$ vectors from the initial input and $K_2, V_2$ vectors from tokens generated via up-interpolation from Scale $Z_1$. During this iteration, the model computes new keys, queries, and values for tokens derived via up-interpolation from Scale $Z_2$, and appends the updated keys/values to the cache for autoregressive generation.

### 3.3 KV-CACHE COMPRESSION IN VAR

Unlike traditional autoregressive Transformers as we mentioned in Definition 3.1, which process tokens one by one in the input stream, the VAR Transformer takes as input tokens from a specific scale token map during each iteration. Specifically, at the start of the $i$-th iteration in the VAR Transformer, its KV cache already contains the key and value vectors from token maps of the first $i - 2$ scales (after up-interpolation operations). During the current iteration, the model will incorporate new key and value vectors from the $(i - 1)$-th scale token maps (after up-interpolation operation) into this cache.

We can formalize the KV cache compression problem in the VAR transformer's setting as follows:

**Definition 3.7** (KV Cache Compress Problem in the VAR Transformer). *At the $i$-th iteration of* VAR *transformer, the model will output the $i$-th scale of feature map $Z_i$ with size $h_i \times w_i$ (See*

*Definition 3.6). In this process, the newly input is a $h_i w_i \times d$-dimension matrix triple $(Q_i, K_i, V_i) \in (\mathbb{R}^{d \times (h_i w_i)})^3$. After the triple entry, the Attention function computes:*

$$\mathsf{Attn}(\mathsf{Q}_i, \mathsf{K}_i, \mathsf{V}_i) := \sigma_i (\mathsf{K}_i \cdot Q_i)^\top \cdot \mathsf{V}_i \in \mathbb{R}^{(h_i w_i) \times d}$$

*where: $\mathsf{K}_i = \begin{bmatrix} K_1^\top & K_2^\top & \cdots & K_i^\top \end{bmatrix}^\top, \mathsf{V}_i = \begin{bmatrix} V_1^\top & V_2^\top & \cdots & V_i^\top \end{bmatrix}^\top$ are $\sum_{j=1}^i (h_j w_j) \times d$ matrices, and $\sigma_i : \mathbb{R}^{\sum_{j=1}^i (h_j w_j) \times h_i w_i} \to \mathbb{R}^{\sum_{j=1}^i (h_j w_j) \times h_i w_i}$ is the softmax function for each column. The $\mathsf{K}_i$ and $\mathsf{V}_i$ matrices are called the key-value (KV) cache.*

For clarity, we present the KV cache update of the VAR transformer during its 3rd iteration in Figure 2.

### 3.4 JOHNSON-LINDENSTRAUSS PROJECTION

In this section, we introduce some concepts about JL-transform from (Woodruff et al., 2014) closely related to our work.

**Definition 3.8** (Definition 3 in (Woodruff et al., 2014), JL-tranform)**.** *For a random matrix $S \in \mathbb{R}^{k \times n}$ forms a Johnson-Lindenstrauss transform with parameters $\epsilon, \delta, f$ or $\mathrm{JLT}(\epsilon, \delta, f)$ for short, if with probability at least $1 - \delta$. then for any $f$-element subset $V \subset \mathbb{R}^n$, for all $v, v' \in V$ it holds that*

$$|\langle Sv, Sv' \rangle - \langle v, v' \rangle| \leq \epsilon \|v\|_2 \|v'\|_2$$

Next, we introduce a lemma that provides a concrete construction of a JL-transform using Gaussian random matrices.

**Lemma 3.9** (Theorem 4 in (Woodruff et al., 2014))**.** *Let $\epsilon, \delta \in (0, 0.1)$ and $S = \frac{1}{\sqrt{k}} R \in \mathbb{R}^{k \times n}$ where the entries $R_{i,j}$ of $R$ are independent standard normal random variables. Then if $k = \Omega(\epsilon^{-2} \log(f/\delta))$, then $S$ is a $\mathrm{JLT}(\epsilon, \delta, \mathrm{f})$*

Next, we present a specific application of the JL-transform in the context of random projections.

**Lemma 3.10** (Johnson-Linderstrauss (JL) Random Projections)**.** *Suppose we have $n$ points $p_1, ..., p_n \in \mathbb{R}^n$ such that $\|p_i\|_2 \leq 1$ for all $i \in [n]$. Let $f : \mathbb{R}^n \to \mathbb{R}^d$ be a random mapping defined as $f(u) = \frac{1}{\sqrt{d}} Au$ where $A \in \mathbb{R}^{d \times n}$ is a random matrix with its entries drawn independently from a standard normal distribution. Then setting: $d \geq \Omega(\epsilon^{-2} \log(n))$ allows us to guarantee that with probability at least $1 - \frac{1}{n}$ it holds for all $(i, j) \in [n]^2$ that $|p_i^\top p_j - f(p_i)^\top f(p_j)| \leq \epsilon$.*

*Proof.* Setting parameters $k = d$ $f = n$ and $\delta = 1/n$, by Lemma 3.9, we have if $d = \Omega(\epsilon^{-2} \log(n))$, then with probability $1 - \frac{1}{n}$, for all $i, j \in [n]^2$, we have $|p_i^\top p_j - f(p_i)^\top f(p_j)| \leq \epsilon \|p_i\|_2 \|p_j\|_2 \leq \epsilon$, where the first step follows from Definition 3.8, and the second step follows from that $\|p_i\|_2 \leq 1$ for all $i \in [n]$. $\square$

## 4 SUB-QUADRATIC SPACE FOR KV CACHE COMPRESSION IS IMPOSSIBLE

In Section 4.1, we introduce the INDEX problem. In Section 4.2, we present the hardness of INDEX problem. In Section 4.3, we extend the INDEX problem to multiple indices. Section 4.4 and Section 4.5 present the communication complexity lower bound for MULTI-INDEX. In Section 4.6, we state the space complexity lower bound for key-value cache in precise attention computation.

### 4.1 INDEX PROBLEM

In this section, we first present the definition of INDEX problem, then we show our main result.

**Definition 4.1** (INDEX Problem,[(Haris & Onak, 2025)])**.** *] Alice holds a bit string $x \in \{0, 1\}^n$ and Bob holds an index $i \in [n]$. Alice sends a single message (one-way) $M \in \{0, 1\}^*$ to Bob, whose goal is to output $x_i$ with probability at least $3/5$.*

## 4.2 Hardness of Index Problem

We present the hardness of INDEX problem which is a well-known result in communication complexity.

**Lemma 4.2** ((Haris & Onak, 2025)). *The one-way, randomized communication complexity of* INDEX *is* $\Omega(n)$.

## 4.3 Multi-Index Problem

Then we extend the INDEX problem, where the index held by Bob changes from 1 to multiple $k$.

**Definition 4.3** (MULTI-INDEX Problem). *Alice holds a bit string* $x \in \{0,1\}^n$, *and Bob holds a subset of $k$ distinct indices* $S = \{i_1, \ldots, i_k\} \subseteq [n]$ *of size $k$. Alice sends a single one-way message* $M \in \{0,1\}^*$ *to Bob. Bob's goal is to output $x_i$ for every index $i \in S$, such that the probability that Bob correctly outputs all $x_i$ where $i \in S$ is at least $3/5$.*

## 4.4 Hardness of Multi-Index Problem

We can extend Theorem 4.2 to the MULTI-INDEX problem.

**Lemma 4.4.** *The one-way, randomized communication complexity of* MULTI-INDEX *is* $\Omega(n)$.

*Proof.* This proof can be easily inducted from Lemma A.2. In particular, it is important to note that when $k = 1$, the MULTI-INDEX problem degenerates into the INDEX problem, in which case the communication complexity lower bound is $\Omega(n)$. When $k \geq 2$, we obviously cannot use less than $\Omega(n)$ bits of information to decode multiple elements held by Alice with the same probability. $\square$

## 4.5 Space Lower Bound Result for Exact Attention Computation

In this section, we provide our lower bound result. In our setting, we assume that the VAR transformer generates a pyramid of feature maps with a total number of scales $N$, where the height and width at the $N$-th scale satisfy $h_N = w_N = n$. Each subsequent level in the pyramid maintains proportional growth in spatial dimensions so that $N = O(\log n)$. The above assumption slightly simplifies the specific implementation in (Tian et al., 2025) while remaining largely consistent. We cover the case of exact computation ($\eta = 0$) on the high dimensional ($d = \Omega(\log n)$) regime to illustrate the key ideas of the proof. We later extend this proof with slightly more complicated arguments to the general case.

We construct a reduction from MULTI-INDEX. Let $L = \sum_{j=1}^{N-1}(h_j w_j)$ denote the total number of tokens up to the $N - 1$-th iteration. Specifically, we can compute that $L := 1^2 + 2^2 + \cdots + (2^{\log n - 1})^2 = \frac{n^2 - 1}{3} = O(n^2)$.

**Theorem 4.5** (Space Complexity Lower Bounds for Key-Value Cache in Precise Attention Computation, formal version of Theorem 1.4). *Let $N$ denote the total number of feature map scales generated by* VAR *transformer, where the height and width at the $N$-th level satisfy $h_N = w_N = n$. There exists some universal constant $C_u > 1$ and for $d \geq C_u \log n$, any algorithm that can, with probability at least $\frac{9}{10}$ produce an output*

$$o_N := \mathsf{Attn}(Q_N, \mathsf{K}_N, \mathsf{V}_N) \in \mathbb{R}^{n^2 \times d}$$

*must use at least $\Omega(n^2 d)$ bits of memory.*

We provide a highly proof idea. We assume Alice holds a bit string $x \in \{0,1\}^{L \times d}$ and Bob holds $n^2$ distinct indices $S = \{i_l, j_l\}_{l=1}^{n^2} \subseteq [L] \times [d]$. Alice sends a single message to Bob, who must output all $x_{i_l,j_l}$ correctly with probability at least $3/5$. By Theorem 4.4, the one-way, randomized communication complexity of this problem is $\Omega(n^2 d)$. Our goal is to design a protocol for MULTI-INDEX by using a supposed algorithm $\mathcal{A}$ for calculating the attention function that uses $S$ bits of memory. Having that reduction in place, Alice simply communicates these $S$ bits of the algorithm's memory tape to Bob, allowing us to show that $S = \Omega(n^2 d)$, and thus proving the theorem.

*Proof of Theorem 4.5.* We provide the formal proof.

**Alice.** Alice begins by inserting the following $L$ triples $\{(q_i, k_i, v_i)\}_{i=1}^{L}$ of vectors in $\mathbb{R}^d$ to the streaming algorithm $\mathcal{A}$:

- $q_1, ..., q_L$ are all the zero vector in $\mathbb{R}^d$, and they do not affect the final output.

- $k_1, ..., k_L \in \mathbb{R}^d$ are calculated before the protocol starts (and agreed upon by both Alice and Bob) as $d$-dimensional projections of the orthonormal basis $e_1 = (1, 0, ..., 0), ..., e_{4n^2} = (0, 0, ..., 1)$ of $\mathbb{R}^{4n^2}$ in a way that approximately preserves orthonormality. Specifically, we can invoke Lemma 3.10 to produce $k_1, ..., k_L \in \mathbb{R}^d$ such that with probability at least $1 - \frac{1}{4n^2}$ it holds for all $i \neq j$ that $|k_i^\top k_j| \leq \epsilon$ and for all $i \in [L]$ that $|k_i^\top k_i - 1| \leq \epsilon$ We do this by letting $f(x) = \frac{1}{\sqrt{d}}Ax$ where $A \in \mathbb{R}^{d \times L}$ is a JL random matrix and defining $k_i = f(e_i)$. Crucially, orthonormality is preserved because $d = \Omega(\log n)$. We resolve the correct value for $\epsilon$ later in the proof.

- $v_1, ..., v_L \in \mathbb{R}^d$ contain rows of $x \in \{0, 1\}^{L \times d}$. In other words, Alice uses $V_n$ to store her input $x$ through $\mathcal{A}$: $\mathsf{V}_n := x$

  After inserting these $L$ triples into $\mathcal{A}$, Alice observes $\mathcal{A}$'s execution and sends its memory state, consisting of $S$ bits, to Bob. This allows Bob to continue the execution of $\mathcal{A}$ exactly where Alice left off, without, of course, having any additional knowledge of $x$.

**Bob.** Recall that Bob's input is an index set $\{(i_l, j_l)\}_{l=1}^{n^2}$ into $x$. In our protocol, Bob will enter a triple $(Q_N, K_N, V_N)$ into $\mathcal{A}$, where he constructs

$$Q_N = C \begin{bmatrix} k_{i_1}^\top & k_{i_2}^\top & \cdots & k_{i_{n^2}}^\top \end{bmatrix}^\top \in \mathbb{R}^{n^2 \times d}$$

$$K_N = \mathbf{0}^{d \times n^2} \quad V_N = \mathbf{0}^{d \times n^2}$$

where $C$ is a positive number. Then we will have $\mathsf{K}_N = \begin{bmatrix} K_1^\top & K_2^\top & \cdots & K_{N-1}^\top & \mathbf{0}^\top \end{bmatrix}^\top, \mathsf{V}_N = \begin{bmatrix} x & \mathbf{0}^\top \end{bmatrix}^\top$ Now, we claim that Bob can recover the value of $\{x_{i_l j_l}\}_{l=1}^{n^2}$ from the output $o_N$, which is equal to $\mathsf{Attn}(Q_N, \mathsf{K}_N, \mathsf{V}_N) \in \mathbb{R}^{n^2 \times d}$. Specifically, we have that

$$\mathsf{Attn}(Q_N, \mathsf{K}_N, \mathsf{V}_N) = \sigma_N(\mathsf{K}_N \cdot Q_N)^\top \cdot \mathsf{V}_N$$

Then we consider for each $l \in [n^2]$, we need to compute

$$(\mathsf{Attn}(Q_N, \mathsf{K}_N, \mathsf{V}_N))_l = \sigma_N(\mathsf{K}_N \cdot Ck_{i_l}^\top)^\top \cdot \mathsf{V}_N$$

Let $s := \mathsf{K}_N \cdot Ck_{i_l}^\top \in \mathbb{R}^{L+n^2}$. And we can have that with probability at least $1 - \frac{1}{L}$ that this is a vector in $\mathbb{R}^{L+n^2}$ with the property that $s_m$ is close to 0 if $m \neq i_l$ and $s_{i_l}$ is close to $C$. Specifically, with probability at least $1 - \frac{1}{L}$:

$$s_m \leq C\epsilon \text{ for } m \neq i_l \text{ and } s_{i_l} \geq C(1 - \epsilon)$$

This is also true vacuously for $m \geq L + 1$ because in this way $s_m = 0$ by our construction. Now, let $\xi := \mathsf{Softmax}(s) \in \mathbb{R}^{L+n^2}$. We can see that the softmax vector $\epsilon$ spikes at index $i_l$. We use this spike to read off $x_{i_l j_l}$ via the $V$ matrix. Let us calculate the product $\epsilon^\top \mathsf{V}_N$. This is a vector in $\mathbb{R}^d$, whose $j_l$-th entry is:

$$(\xi^\top \cdot \mathsf{V}_N)_{j_l} = \sum_{m=1}^{L} x_{m j_l} \xi_m$$

$$= x_{i_l j_l} \xi_{i_l} + \sum_{m \neq i_l} x_{m j_l} \xi_m$$

We can now examine two separate cases: **Case 1:** $x_{i_l j_l} = 0$. Then we have that

$$(\xi^\top \cdot \mathsf{V}_N)_{j_l} = \sum_{m \neq i_l} x_{m j_l} \xi_m$$

$$= \frac{\sum_{m \neq i_l} x_{m j_l} e^{s_m}}{e^{s_{i_l}} + \sum_{m \neq i_l} e^{s_m}}$$

$$\leq \frac{\sum_{m \neq i_l} e^{s_m}}{e^{s_{i_l}} + \sum_{m \neq i_l} e^{s_m}}$$

The function $\frac{x}{x+y}$ is maximized when $x$ is maximized and $y$ is minimized, which allows us to bound:

$$(\xi^\top \cdot \mathsf{V}_N)_{j_l} \leq \frac{(L + n^2 - 1) e^{C\epsilon}}{(L + n^2 - 1) e^{C\epsilon} + e^{C(1-\epsilon)}} := \delta$$

**Case 2:** $x_{i_l j_l} = 1$. Then we have that

$$(\xi^\top \cdot \mathsf{V}_N)_{j_l} = x_{i_l j_l} \xi_{i_l} + \sum_{m \neq i_l} x_{m j_l} \xi_m$$

$$= \frac{e^{s_{i_l}} + \sum_{m \neq i_l} x_{m j_l} e^{s_m}}{e^{s_{i_l}} + \sum_{m \neq i_l} e^{s_m}}$$

$$\geq \frac{e^{s_{i_l}}}{e^{s_{i_l}} + \sum_{m \neq i_l} e^{s_m}}$$

The function $\frac{x}{x+y}$ is minimized when $x$ is minimized and $y$ is maximized, which allows us to bound:

$$(\xi^\top \cdot \mathsf{V}_N)_{j_l} \geq \frac{e^{C(1-\epsilon)}}{(L + n^2 - 1) e^{C\epsilon} + e^{C(1-\epsilon)}} := \Delta$$

For Bob to always be able to distinguish between the two cases, we want to ensure that $\delta \leq \Delta$. Then we can choose $C = \frac{2 \ln(L + n^2 - 1)}{1 - \epsilon}$ and $\epsilon = 0.1$ to allow Bob to distinguish between $x_{i_l j_l} = 1$ and $x_{i_l j_l} = 0$ with probability at least $1 - 1/4n^2$. Then for all $l \in [n^2]$, the probability of the event that Bob can correctly distinguish all $\{x_{i_l, j_l}\}$ is $(1 - 1/4n^2)^{n^2} \geq 3/5$ when $n \geq 2$. Then we conclude the proof. $\square$

### 4.6 Space Lower Bound Result for Approximate Attention Computation

Now we extend the above result to the approximate computation of the attention function.

**Theorem 4.6.** *Let $Z_N := \mathsf{Attn}(Q_N, \mathsf{K}_N, \mathsf{V}_N)$ and $d = \Omega(\log n)$. Any algorithm that can, with probability at least $9/10$ produce an output $\mathcal{O} \in \mathbb{R}^{n^2 \times d}$ that is a $(1 \pm \eta)$-approximation of $Z_n$ for $\eta \in (0, 1)$ must use at least $\Omega(n^2 d)$ bits of memory.*

*Proof.* Our reduction follows the same approach as before, except that Bob now uses $\mathcal{O}$ to distinguish between the cases $x_{i_l j_l} = 0$ and $x_{i_l j_l} = 1$. Specifically, when $x_{i_l j_l} = 0$, we have $\mathcal{O}_{j_l} \leq (1 + \eta)\delta$, whereas for $x_{i_l j_l} = 1$, it holds that $\mathcal{O}_{j_l} \geq (1 - \eta)\Delta$, where $\delta$ and $\Delta$ are as defined in Theorem 4.5.

To ensure distinguishability, we require $(1 + \eta)\delta < (1 - \eta)\Delta$, which simplifies to $\delta < \frac{1-\eta}{1+\eta}\Delta$. This leads to the bound $C = \Omega(\ln n - \ln \frac{1-\eta}{1+\eta})$, accounting for $\epsilon = 0.1$. The proof is similar to the one above, and we omit it for brevity. $\square$

## 5 Conclusion

This work establishes foundational theoretical insights into the memory complexity of KV-cache compression for Visual Autoregressive transformers. We rigorously formalize the KV-cache compression problem and prove that any attention-based mechanism for sequential visual token generation inherently requires $\Omega(n^2 d)$ memory under standard architectural assumptions, where $n$ denotes the height and width of the last scale token map generated by Visual Autoregressive transformer and $d$ is the embedding dimension. This result demonstrates the impossibility of achieving sub-quadratic memory consumption without introducing additional structural constraints or approximations.

## ETHIC STATEMENT

This paper does not involve human subjects, personally identifiable data, or sensitive applications. We do not foresee direct ethical risks. We follow the ICLR Code of Ethics and affirm that all aspects of this research comply with the principles of fairness, transparency, and integrity.

## REPRODUCIBILITY STATEMENT

We ensure reproducibility of our theoretical results by including all formal assumptions, definitions, and complete proofs in the appendix. The main text states each theorem clearly and refers to the detailed proofs. No external data or software is required.

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

# Appendix

## LLM USAGE DISCLOSURE

LLMs were used only to polish language, such as grammar and wording. These models did not contribute to idea creation or writing, and the authors take full responsibility for this paper's content.

