# OpenReview forum: "Visual Autoregressive Transformers Must Use $\Omega(n^2 d)$ Memory"
_ICLR.cc/2026/Conference — Submitted to ICLR 2026_

### Official Review · Reviewer_qmvb · 2025-10-19

**Soundness:** 2
**Presentation:** 4
**Contribution:** 2
**Rating:** 2
**Confidence:** 4

**Summary:**

The authors propose a theoretical framework for Visual Autoregressive (VAR) Transformers, aiming to formally prove a fundamental lower bound on KV-cache memory. They rigorously define the KV-cache compression problem (Def. 3.7) and prove through a reduction from the MULTI-INDEX communication complexity problem that any attention-based VAR architecture must use at least Ω(n²d) memory when
d=Ω(logn). In plain terms, the paper argues that no matter how cleverly you compress or sparsify, sub-quadratic memory scaling is impossible without changing the model’s structure or approximation behavior. The proof leverages the Johnson–Lindenstrauss projection to ensure orthonormal preservation of key embeddings and builds a clean reduction argument showing that exact or even approximate attention computation can’t break the Ω(n²d) barrier.

**Strengths:**

The derivation of the lower bound (Theorem 4.5 – lines 354–377) is impressively formal. By reducing the VAR attention mechanism to a communication-complexity problem (INDEX → MULTI-INDEX → VAR KV), the authors give a clear and elegant chain of logic linking information theory and transformer memory scaling. It’s rare to see such a well-structured mathematical treatment of a practical deep-learning bottleneck.

The work defines the KV-cache compression problem precisely for VAR Transformers (Defs 3.6–3.7, lines 268–323), filling a theoretical gap in prior LLM-style KV-compression studies. This clarity makes it a valuable foundation for future research that might seek looser assumptions or approximate formulations.

**Weaknesses:**

Static-size assumption challenged by modern dynamic KV methods: The entire theorem assumes fixed-size, full-precision KV caches across iterations, but recent works introduce dynamic cache resizing, token reshaping, and low-rank compression (e.g., dynamic eviction, adaptive rank KV, or reparameterized state-space compression). These violate the paper’s fixed-architecture assumption and could, in principle, sidestep the Ω(n²d) bound. Thus, while the theorem is mathematically valid, its real-world applicability is limited.

Simplifying assumptions weaken generality: The reduction proof assumes d=Ω(logn) and strictly orthogonal key embeddings (lines 388–412). Real visual tokens are highly correlated, so these assumptions might not hold. The resulting lower bound might therefore describe an idealized scenario rather than realistic visual generation systems.

No discussion of approximate or structured attention breakthroughs: Although Theorem 4.6 mentions approximate attention, the paper doesn’t analyze how structured sparsity (block, low-rank, or locality-aware) could loosen the bound. Dynamic models like LongRoPE, SVD-based KV, or FlashAttention-v3 partially overcome quadratic scaling, but these aren’t discussed.

**Questions:**

Could this Ω(n²d) lower bound still hold if KV tensors were dynamically reshaped or compressed (e.g., rank-adaptive projection or sliding-window attention)?

Is there any pathway—perhaps via approximate random-feature attention or structured pruning—that might asymptotically reduce the bound without breaking attention accuracy guarantees?

---

### Official Review · Reviewer_cWWG · 2025-10-21

**Soundness:** 3
**Presentation:** 1
**Contribution:** 2
**Rating:** 4
**Confidence:** 2

**Summary:**

This paper investigates the theoretical memory complexity of Visual Autoregressive (VAR) Transformers, particularly focusing on the KV-cache compression problem during autoregressive image generation. The authors first formally define the KV-cache compression problem in the VAR setting and establish a space complexity lower bound showing that any exact or approximate attention-based generation mechanism must require Ω(n²d) memory. The proof constructs a reduction from the MULTI-INDEX problem in communication complexity, employing Johnson–Lindenstrauss embeddings to connect random projections with orthonormal preservation. The result demonstrates the impossibility of achieving sub-quadratic KV-cache memory without imposing additional structural constraints.

**Strengths:**

- The paper provides a formal definition of the KV-cache compression problem in a visual autoregressive context, and the formalization bridges a conceptual gap between text autoregressive transformers and image-based variants.
- The paper establishes a strong theoretical contribution and a lower-bound proof, exhibiting a clear theoretical roadmap and structure.

**Weaknesses:**

- No experiments or numerical simulations are presented to corroborate the theoretical bounds. It may be beneficial if it is possible to demonstrate whether practical KV-cache usage in real VAR models (e.g., Tian et al., 2025) approaches the theoretical limit. Without empirical results, it remains unclear how tight the bound is in real-world inference scenarios.
- Limited discussion on model assumptions. No discussion is provided on how sparsity priors or approximate attention mechanisms (e.g., low-rank or clustered attention) could potentially evade the bound, although mentioned briefly in the abstract. The assumptions of uniform scaling and independence across scales (Sec. 4.5) may oversimplify real VAR architectures.
- Related work primarily cites LLM-based KV compression. The paper could better situate its contribution within recent theoretical analyses of diffusion or autoregressive complexity.
- No summary figure or pseudocode is given for the proof construction (only algebraic exposition).

**Questions:**

see weaknesses

---

### Official Review · Reviewer_YtG6 · 2025-10-28

**Soundness:** 1
**Presentation:** 1
**Contribution:** 2
**Rating:** 2
**Confidence:** 4

**Summary:**

This paper formalizes the KV-cache compression problem for visual autoregressive models and establishes a lower bound of $\Omega(n^2d)$ memory via reductions to communication complexity and Johnson–Lindenstrauss projections. The main result claims that sub-quadratic memory is impossible for exact or approximate attention in VAR models.

**Strengths:**

- Addresses a highly relevant problem in efficient inference for VAR Transformers, clarifying theoretical memory limits.

- Covers both exact and approximate attention, making the results broadly applicable.

**Weaknesses:**

- There is strong evidence of substantial overlap with another submission (Submission 13283), including nearly identical figures and technical content, making this paper appear to be either a duplicate submission or written by an LLM.

- The formalization lacks rigor in notation and dimension consistency, and key technical assumptions (e.g., JL projection applicability) are not well justified.

- Figures and formulas suffer from severe formatting issues (garbled symbols, unreadable images), which hinder understanding.

**Questions:**

- Please clarify the dimension consistency and normalization in your attention definitions, ensuring all matrix operations are valid.

- How can the JL projection assumptions be realized in practical VAR pipelines? Are there concrete mechanisms or counterexamples?

---

### Official Review · Reviewer_ewrz · 2025-10-29

**Soundness:** 2
**Presentation:** 2
**Contribution:** 2
**Rating:** 2
**Confidence:** 3

**Summary:**

The paper formally defines the KV-cache compression problem for Visual Autoregressive (VAR) Transformers and proves a fundamental lower bound: any algorithm that computes attention (exactly or within a constant approximation) must use at least $\Omega(n^2 d)$ memory, where $n^2$ is the number of tokens at the final image scale and $d$ is the embedding dimension (with $d = \Omega(\log n)$). This implies that sub-quadratic memory usage is impossible without additional structural assumptions such as sparsity.

**Strengths:**

This paper establishes the first rigorous theoretical framework for KV cache compression in visual autoregressive models, proving the impossibility of sub-quadratic memory under standard assumptions.

**Weaknesses:**

1. The theoretical analysis relies on worst-case assumptions, such as nearly orthogonal and random keys, which may not reflect the structured and locally correlated nature of real visual data.

2. The paper lacks empirical evaluation or simulation to compare its theoretical lower bound with the actual memory-performance trade-offs of existing KV cache compression techniques in visual autoregressive models.

3. Figure 1 in this paper is identical to figure 1 in another submission (https://openreview.net/forum?id=FriP8PXHpY).

**Questions:**

See weakness.

---

### Meta-Review · Area_Chair_3Qd6 · 2026-01-18

**Summary:**

This paper overlapped too much with "CIRCUIT COMPLEXITY BOUNDS FOR VISUAL AUTOREGRESSIVE MODEL", another submission at ICLR 2026. The reviewers also raised many concerns and reached a consensus to reject the paper, but the authors did not provide a rebuttal.

**Reviewer Concerns:**

No concerns addressed as no rebuttal provided.

**Reviewer Scores:**

No change as no rebuttal provided.

---

### Decision · Program_Chairs · 2026-01-26

Reject